# Consumption in the Circular Economy: Learning from Our Mistakes

**Dimitris Georgantzis Garcia** [1,2,*] **, Eva Kipnis** [1]**, Efi Vasileiou** [3,4] **and Adrian Solomon** [2]

1. Sheffield University Management School, University of Sheffield, Sheffield S10 1FL, UK; eva.kipnis@sheffield.ac.uk
2. South East European Research Centre (SEERC), 546 22 Thessaloniki, Greece; asolomon@seerc.org
3. GREDEG, Université Côte d'Azur, 06560 Valbonne, France; evasileiou@york.citycollege.eu
4. CITY College, University of York Europe Campus, 546 26 Thessaloniki, Greece
* Correspondence: dgarcia@seerc.org or dgeorgantzisgarcia1@sheffield.ac.uk

**Abstract:** The Circular Economy (CE) is gaining increasing attention among businesses, policymakers and academia, and across research disciplines. While the concept's strong diffusion may be considered its main strength, it has also contributed to the emergence of many different understandings and definitions, which may hinder or slow down its success. Specifically, despite growing attention, the role of the consumption side in the CE remains a largely under-researched topic. In the present review, we first search the literature by means of snowball mapping and a systematic key-word strategy, and then critically analyze the identified sources in order to elucidate the fundamental elements that should characterize consumption in a CE. We extract two pillars, directly from definition, that should be at the nucleus of future research on consumption in the CE: (1) the hierarchical nature of circular strategies, with "reduce" being preferred to all other strategies; and (2) the inadequacy of defining the CE only through its loops or strategies without considering its goal of attaining sustainable development. Moreover, the discussion is placed within the extant consumer research streams deemed relevant, in order to bridge these with the context of the CE. We highlight limitations of said research streams regarding their typical focus on the quality (and not the quantity) of consumption, the lack of heterogeneity in the theories and data collection methods employed, and the non-impact-based instruments typically used to measure consumption behaviors. We show how these limitations have contributed to the emergence of the intention–behavior gap, a phenomenon extant studies identify as key to overcome for encouraging sustainable consumption practices. In particular, we focus the analysis on the intention–behavior gap in order to: (1) establish the state-of-the-art; and (2) uncover avenues for future research addressing extant limitations.

**Keywords:** circular economy; sustainable consumption; consumer behavior; sustainable development; intention–behavior gap; planetary boundaries

## 1. Introduction

Current consumption levels in affluent nations are unsustainable and account for an important share of the overall negative environmental impacts caused by human activity. However, consumption patterns cannot change enough to overcome this problem in an economic context which incentivizes the constant creation and fulfilment of needs [1,2]. Hence, calls for a socio-economic development paradigm shift, towards a sustainable one, are becoming increasingly urgent as human activity continues to push on the Earth system's (ES) boundaries [3,4]. Planetary boundaries have been defined on the ES [3], beyond these, the impacts of human–economic activity on the environment may result in unpredictable and potentially irreversible changes, posing a significant threat to the possibility for overall human life on the planet. Subsequently, [4] provide the latest advancements within this framework showing that at least four, of the seven currently measurable boundaries, have already been exceeded. Consequently, the need to work towards a system

of socio-economic development that operates within the ES's planetary boundaries is a general priority and requires a system capable of facilitating the adoption of sustainable consumption patterns.

The Circular Economy (CE) represents the most current attempt at addressing the aforementioned risks by re-thinking the business models that drive our societies from all possible perspectives, including consumption. The CE is restorative by intention and aims to achieve sustainable development by optimizing the socio-environmental system as a whole, and not just its individual components, through a complex systems perspective [5,6]. Moreover, the CE's material flows require that biological nutrients are safely fed back to the biosphere while technical nutrients should be re-circulated, maintaining their quality, without entering the biosphere (i.e., industrial ecology) [6]. Essentially, it aims to integrate human–economic activity and the natural environment in a sustainable manner [7] and is currently being widely promoted by several nations, international bodies (e.g., China, Japan, UK, France, Netherlands, Spain) and businesses around the world (e.g., Danone and Patagonia) [8]. This is not surprising as sustainable development continues to gain increasing attention among policymakers, international organizations [1,9,10], and academics [4,11,12] alike.

Since the rise of neoliberal capitalism in the 1970s, the world has seen an overwhelming growth in global economic activity and in life standards [13]. These outcomes are only possible when accompanied by the constant creation of new consumer needs (or desires) which the market then seeks to accommodate. However, time is showing that the continuous growth in consumption embraced by the current economic context, and the consequential resource exploitation, collide with the Earth's planetary boundaries [3,4]. As a result, environmental policy debates internationally have been dominated by the idea that consumption currently represents one of the main causes of the growing environmental problems [9,14,15], if not the main one [1]. Moreover, the CE is characterized by initiatives for a better use of resources and waste management whose success often relies on the acceptance and/or active engagement of end consumers. Their behavior and decision-making at purchase, use and end-of-life management stages of goods can enable or hinder the success of the CE's initiatives [16,17]. These two arguments place consumption at the core of ensuring the CE's success, not only in terms of consumers' acceptance and adoption of CE-initiatives, but also with respect to the CE's ultimate goal of attaining sustainable development [5]. In sum, the role of consumption in the CE is of interest to all policymakers, academics, industry and civil society, especially given that these stakeholders are all equally embedded in both society and the environment.

There have been relevant efforts to elucidate environment–attitudinal factors and mechanisms behind purchasing behavior of consumers, relative to products differentiated through their reported circular attributes [18]. Moreover, [19] explore and comprehensively taxonomize the material flows that characterize the CE by considering examples of designs aiming to recirculate products at their end-of-life stage. In doing so, the authors identify valuable practical challenges to the implementation of the CE. Despite these nascent developments, the consumption side has not been given nearly as much attention as the production side in the CE literature. In particular, a clear characterization of consumption in the CE, drawn from the growing academic literature on CE and extant literatures that overlap with the latter, by means of a critical review, is currently missing from the literature. As a result, factors linked to consumers' habitual consumption and consumption culture and means for enabling the success of a CE by focusing on its goals remain poorly understood [16,17]. The study in [17] begins to address this problem by offering a valuable summarizing account of existing research on the topic of consumption in the CE. Extending this effort, the present research aims to highlight related streams of research related to CE (e.g., sustainable or ethical consumption) while pointing out several commonly overlooked limitations that can be imposed by the direct transfer of these concepts to CE-focused studies. Our review aims to fill this knowledge gap by synthesizing and critically reviewing the literature, in an attempt to build a sound conceptual foundation to help set a coherent direction for future research related to consumption in the CE. In this

regard, our review was guided by the following research question: *How should consumption in the CE be characterized and how does its characterization relate to extant research streams and sustainable development?*

Partly due to its interdisciplinary nature, research on the CE is currently lacking a formal academic debate in the social sciences and sustainability literature, making the former sparse and often confusing [7]. The challenging task of developing a definition of sufficient specificity and rigor capable of capturing all the characterizing elements and goals of the CE has also contributed to said phenomenon [5]. In fact, despite the existence of an overwhelming number of definitions of the CE concept, new definitions continue to emerge with different foci and priorities [5,11,20]. This results in heterogeneity in extant understandings of the concept across different disciplines and societal actors (academia, business, government etc.) [11], which may hinder its potential to succeed as a strategy towards sustainable development. Based on the definition developed by [5], the present review identifies sustainable consumption to be one of the micro-level foundations of the CE. Furthermore, this point is utilized to construct a conceptualization of consumption in the CE, in relation to existing streams of research, a critical step in exploring the factors and mechanisms behind the pertinent consumption behaviors. In order to provide the foundation for the rest of this review, these conceptualizations are addressed and discussed in Section 3.

The identification of sustainable consumption as a micro-level basis of the CE implies that, while the CE as such is a new concept, the consumption behaviors it entails can often be fitted within existing streams of research. More specifically, due to the changing definitions of sustainable consumption (and other similar concepts), from one study to the next, the present review explores the significant overlap between sustainable, responsible, ethical, green and pro-environmental consumption (see Section 3). These research streams tend to focus on the formation of behavioral intentions (BI) and attitudes that are then assumed to significantly predict behavior [21–24]. Consequently, the scope of the conclusions to be drawn about behavior from this stream of research is limited by a phenomenon known as the attitude–behavior gap or *intention–behavior gap* (IBG) [25]. This refers to the widely reported misalignment between consumers' reported attitudes/intentions and their actual behavior. For example, most Europeans report that they engage in waste management practices currently and are willing to engage with new business models [26]. However, these claims do not match observations of behavior in the real world [16]. While commonly reported in ethical consumption contexts relevant to the CE, the IBG is still poorly understood [16,17]. Yet, the IBG is critical to take into account and address in CE studies given its prominent impact identified in the related literature streams.

The current free-market economic systems imply unsustainable consumption patterns and consumerist cultures, even when including green alternatives [2,27,28]. As a result, although useful, it is not sufficient to explore consumer preferences for sustainable alternatives of products/services (i.e., the quality of consumption). The quantity of consumption plays a crucial role in making consumption sustainable [2]. Hence, a deeper cultural shift is necessary, capable of driving the emergence of eventually sustainable consumer lifestyles and consumption patterns [29,30]. Despite that, the quantity of consumption and the factors behind it, such as culture and institutional context, are currently underexplored topics [29]. In sum, within the context of sustainable development and CE, there is an urgent need to understand consumption and, more specifically, the quantity of consumption. Moreover, understanding the mechanisms behind the formation of the IBG can help elucidate the main barriers that prevent even the responsible consumer from acting in coherence with their BI or attitudes.

The section that follows reports the methods employed in the literature identification stage of the present review, as well as the reasoning behind the choice of the more qualitative, critical approach to the analysis of the sources. Section 3 offers a discussion on the definition of the CE extracting core elements for understanding the role of consumption. Additionally, the extracted concepts are placed against existing literatures that study

pertinent consumption behaviors. Section 4 seeks to illustrate the current unsustainability of human-originated perturbation of the ES. Moreover, the importance of combining bottom-up and top-down CE-enabling initiatives is stressed, together with the importance of the reduce strategy included in the definition of the CE and its relation to the concept of sufficiency. Section 5 begins by offering a small discussion based on an existing review of instruments for measuring sustainable consumption behavior (SCB) through self-reported data, highlighting the particularly relevant nature of impact-based measures of behavior. The section then continues by reviewing the literature pertinent to the IBG and the CE and key weaknesses to watch out for in future research are identified. Finally, Section 6 offers a summary and discussion of the key points and findings of the review and its main contributions to the literature. The latter section then concludes the review by providing a discussion of the main managerial implications, limitations of our work and avenues for future research. A summary of the key themes covered in the review, broken down into main topics and subsequent key points, together with their location within the review and the key associated references, is presented in Table 1.

**Table 1.** Review navigation table: summary of key points and gaps identified.

| Theme | Topics | Key Points/Gaps | Sec. | Key References |
|---|---|---|---|---|
| Characterization of consumption | Pillars of consumption in the Circular Economy | (1) Hierarchy of circular strategies (i.e., preference for the reduce strategy), (2) inadequacy of defining CE-consumption without regard for its goals. | 3 | [5,11,17,19,31] |
|  | Overlaps and fundamental differences with extant concepts | Focus on consumption drivers (i.e., sustainable, ethical, responsible, green and pro-environmental consumption) vs. Product-as-Service Systems, collaborative consumption or stakeholder involvement. |  | [32–41] |
| The reduce strategy in consumption | Population, affluence and technology as determinants of global sustainability | The importance of considerations of sufficiency (quantity of consumption) vs. green consumerism (quality of consumption), currently underrepresented by the literature. | 4 | [1–4,27–30,42–57] |
|  | A neo-institutional perspective on the sustainability of consumption | A combination of bottom-up and top-down strategies necessary for a successful transition towards sustainable development. Shared citizen–government responsibility. |  | [29,30,47,58–82] |
| The intention–behavior gap (IBG) in sustainable consumption | Self-report measures of sustainable consumer behavior (i.e., methodologist perspective on the IBG) | Employing considerations of pro-environmental consumption tendencies, diary procedures or impact-based measures. | 5 | [83–85] |
|  |  | Need for innovation in data-collection methods where incentive compatibility criteria are met. |  | [86–99] |
|  | Theory of Planned Behavior widely favored (i.e., modeler perspective on the IBG) | Research tends to focus on the formation of behavioral intentions under the assumption that they will strongly predict behavior. |  | [17,24,25,86–102] |
|  |  | Need for innovation regarding the conceptual models employed and empirical testing of extant conceptualizations. |  | [25,90–118] |

## 2. Materials and Methods

The literature review presented in Sections 1–4 first identified a basis of recent literature review articles (last 4 years) through a keyword search, on Scopus, Web of Science and

Google Scholar. This contained terms relevant to consumption and the CE in general and low-impact journals (i.e., journals rated lower than 3 or B in the Charted Association of Business Schools or the Australian Business Deans Council, respectively) were excluded. The latter criterion did not affect subject-specific (sustainability, CE and so on) journals and all articles were inspected in further depth for inclusion/exclusion. Having identified these basis articles, snowballing techniques [119] were employed to find the next set of relevant articles from the basis articles' references. The process was then iterated to identify further sources. This snowballing approach was particularly useful for identifying studies that use different vocabulary to refer to the same (or similar) concepts, which can be a problem when searching for a given set of keywords [119].

In Section 5 the literature search followed a systematic keyword search (see Table 2), in order to identify all relevant literature to the IBG in sustainable consumption and the CE. This method was deemed more suitable here as the narrow scope of the section's focus restricts the use of incoherent vocabulary across studies. Additionally, the method's benefits in terms of rigor and replicability are also desirable [120]. The strategy (shown in Table 2) consisted of two stages that were constructed based on the results of the first part of the review, where overlapping streams of research that align with the concept of consumption in the CE were identified, as outlined in Section 3. Furthermore, the terms were searched for in the titles, abstracts and key words of academic papers (both in journals and books) from 2010 onwards by using the Scopus and Web of Science databases. The articles were recorded on 17 June 2020 (n = 151). Duplicates emergent from both databases were removed (n = 93), low-impact journals (i.e., journals rated lower than 3 or B in the Charted Association of Business Schools or the Australian Business Deans Council, respectively) were excluded without affecting subject-specific (sustainability and CE) journals; abstracts were inspected in order to identify the relevant papers for inclusion. Articles whose primary focus was not the exploration or elucidation of elements pertaining to consumption behavior and psychology in contexts implying ethical or sustainability considerations were excluded in this inspection (n = 77). Finally, after full-text screening remaining articles on the latter criterion, additional articles were excluded while particularly relevant pieces of work identified in the reference lists of the analyzed articles were included in the review for completeness. The final sample drawn for analysis comprised n = 56.

**Table 2.** Literature search strategy: The sustainable consumption intention–behavior gap (IBG).

| Stage 1: How is the attitude–behavior (intentions–behavior) gap understood in the existing literature? |
| --- |
| "attitud*-behavi*r gap" OR "intentio*-behavi*r gap" OR "attitud* behavi*r gap" OR "intentio* behavi*r gap" OR "hypothetica* bia*" |
| Stage 2: How does the attitude–behavior (intentions–behavior) gap currently relate to the topic of sustainable/ethical/green consumption or in the CE? |
| AND<br>"green consum*" OR "sustain* consum*" OR "ethic* consum*" OR "circular economy" OR "circula* consum*" OR "ecol* preferenc*" OR "CSR" OR "corporate social responsibility" OR "responsible consum*" OR "conscious consum*" OR "pro-environmental consum*" OR "environmental* consum*" |

The literature identified through the methods described above was subjected to a critical review [120] in order to identify the current state of the topic and extant research gaps that need to be addressed. The benefits of snowball mapping approaches and a key-word literature search strategy, drawn from typical methods in systematic reviews, are desirable in order to maximize the reportability and replicability of the literature identification stage of the present work. However, rather than a descriptive or quantitative analysis of the identified literature, this review seeks to offer a "diagnostic" of extant research relevant to sustainable consumption and lay down avenues for future research capable of addressing certain shortcomings while correctly characterizing consumption

in the CE. As [120] (p. 93) explains, "a critical review provides an opportunity to 'take stock' and evaluate what is of value from the previous body of work". Therefore, a critical approach provides the best perspective for our purposes by offering qualitative insights on the achievements and pending explorations of the existing literature [120].

### 3. Consumption in the Circular Economy: The Predecessors

The diffusion of the CE concept into numerous research disciplines and the interest it awakens in academia, businesses and governments alike, is arguably the main factor differentiating it from previous attempts moving towards similar directions. In fact, this can be understood as its main strength since it significantly increases the probability of successful prospects with regards to its practical application. However, this interest for the CE shared among actors and disciplines has also led to the creation of many different understandings of the concept, making its application and research often incompatible from one perspective to the next. A direct consequence of these differing views is the emergence of an overwhelming number of different definitions of the CE in the academic literature.

In their work, the authors of [5] draw from 114 definitions of the CE in order to identify essential elements that characterize the CE. They then develop the following definition, which is the one adopted by the present review, by combining all characterizing elements of the CE: "A Circular Economy describes an economic system that is based on business models which replace the "end-of-life" concept with reducing, alternatively reusing, recycling and recovering materials in production/distribution and consumption processes, thus operating at the micro level (products, companies, consumers), meso level (eco-industrial parks) and macro level (city, region, nation and beyond), with the aim to accomplish sustainable development, which implies creating environmental quality, economic prosperity and social equity, to the benefit of current and future generations" (pp. 224–225). This definition was strategically put together by the authors in order to capture the most critical elements of other, less holistic definitions that had been previously developed. In particular, two elements are particularly relevant to considerations about consumption:

1.  The order of the circular-loop strategies in the definition is not arbitrary. Instead, it is hierarchical, meaning that reducing should be preferred to reusing or any other strategy appearing later on in the definition, and so on [5,19]. This is also highlighted by Europe's waste-management hierarchy [17,19] and movements like the Zero Waste International Alliance who aim to boost the adoption of waste management strategies in line with the CE through education, recommendations for policy and benchmarking [31].
2.  When defining the CE, it is not sufficient to include a definition of the strategies or initiatives that characterize the CE without defining the goals of said strategies [5,11]. Therefore, by definition, processes that may appear to be circular in the way they operate are not necessarily circular if they do not contribute to overall sustainable development in all its dimensions (environmental, social and economic).

It is important at this point to understand the meaning of "(un)sustainable development". This literally refers to socio-economic development that can (or cannot) be sustained over time and space. A system may be very effective in driving societal development. However, if it cannot be sustained, there is a limit to how long the system can operate for, resulting in the system's unsuitability as a strategy for the attainment of long-term or world-wide societal development. Hence, it does not suffice to have a good economic system if it is not also sustainable. Current socio-economic systems require resource extraction from the ES and imply technological and other processes, all of which put pressure on the ES. Within this line of thought, as illustrated in Sections 1 and 4, the current paradigm of socio-economic development is not sustainable [3,4]. Consequently, important efforts such as the Sustainable Development Goals (SDGs) of the United Nations [121] have emerged, aiming to address key flaws of the socio-economic system with respect to its lack of sustainability.

As Peattie [32] explains, consumption behaviors that aim to achieve sustainable development reflect the United Nations Environmental Program's [33] notion of sustainable consumption (see Table 3), so, based on the above-mentioned definition of the CE, sustainable consumption constitutes part of the micro-level basis on which the CE paradigm is built. Therefore, a successful transition towards a CE requires a change of consumption patterns such that they become increasingly sustainable (i.e., in coherence with sustainable development). Although the CE may be considered a new concept, the consumption behaviors it entails (i.e., SCB) have been researched and documented to a reasonable extent. Moreover, the concepts of "ethical consumer behavior", "green consumption", "responsible consumer behavior" and "pro-environmental consumer behavior" (PECB) overlap significantly with that of SCB. This creates ambiguity and a lack of consistency regarding their conceptions in the literature [32]. Table 3 shows a pair of "insights" (definitions, perspectives or similar), about the four key concepts listed previously, that illustrate the inconsistencies in their defining foci. A more detailed discussion on these follows.

**Table 3.** Consumption in the Circular Economy (CE): Coherent existing concepts.

| Concept | Insight 1 (I1) | Insight 2 (I2) |
|---|---|---|
| Sustainable consumption | "actions that result in decreases in adverse environmental impacts as well as decreased utilization of natural resources across the lifecycle of the product, behavior, or service. [ ... ] improving environmental sustainability can result in both social and economic advances" [34] (p. 24) | "a number of key issues, such as meeting needs, enhancing quality of life, improving efficiency, minimizing waste, taking a life cycle perspective and taking into account the equity dimension, for both current and future generations, while continually reducing environmental damage and the risk to human health" [33] |
| Green consumption | "as shorthand for oriented toward sustainable development. This reflects the United Nations Environment Program's conception of sustainable consumption" [32] (p. 197) | Green consumers defined "as those individuals who engage in a set of pro-environmental behaviors (e.g., recycling, reducing household waste) primarily for environmental reasons" [35] (p. 230) |
| Ethical consumption | "consumption activities that are consistent with conscience, values, and morals" [37] (p. 507) | "the purchase of a product that concerns a certain ethical issue and that a consumer chooses freely" [38] (p. 512) |
| Responsible consumption | "in 2015, the United Nations introduced a new series of goals called 'Sustainable Development Goals' (SDGs) made up of 17 goals and 169 associated targets to be achieved over the next 15-year period starting from 2016 until 2030" [121] (p. 2) "Responsible Consumption and Production is the twelfth SDG goal" [121] (p. 6) | This is also given the alternative name "Goal 12: Ensure sustainable consumption and production patterns" in official UN documents. In other words, the concepts of "responsible" and "sustainable" consumption are understood to have the exact same meaning. |

In the case of sustainable consumption (or SCB), insights 1 and 2 (I1 and I2 in Table 3) offer two definitions of the concept drawn from two different but equally rigorous sources. In the first (I1), a clear preference in focus is given to the environmental dimension of sustainability: "actions that result in decreases in adverse environmental impacts [...] de-

creased utilization of natural resources..." [34] (p. 24). As shown in the table, the authors go on to justify this preference by arguing that the dimensions of sustainability are in fact non-orthogonal (i.e., they depend on one another) and that the environmental dimension has significant potential to improve the other two [34]. On the other hand, the second definition (I2) quite clearly addresses all three dimensions of sustainability equally: "enhancing quality of life"; "continually reducing environmental damage"; "meeting needs [...] for both current and future generations" [33]. This definition also highlights the future-looking perspective of sustainability, which is not explicitly addressed by [34], further illustrating their differences.

For green consumption (second row of Table 3), I1 provides a definition which the author arrives at following an insightful discussion on the lack of consensus regarding the definitions of these concepts: "as shorthand for oriented toward sustainable development" [32] (p. 197). This definition essentially results in the use of "green" to mean "sustainable", clearly illustrating the overlap. On the other hand, I2 provides a definition of "green consumers" that focuses on the pro-environmental nature of the behaviors, "individuals who engage in a set of pro-environmental behaviors" and the reasons for their action, "primarily for environmental reasons" [35] (p. 230). Therefore, the focus could be understood as more environmental than social or economic, in which case, these two conceptions of green consumption lack coherence. However, the meaning of PECB, has been shifting towards that of SCB in recent years as can be seen in the work by [32,122–125]. If this was to be taken into account in the comparison of I1 and I2, then the overlap between "green" and "sustainable" becomes even more clear.

At this point, it is worth mentioning that I2 for green consumption (Table 3) is in line with the notion of "spillover effects" [36], whereby a consumer's PECB in one setting can spill over to other settings if the reason for behaving sustainably is mostly environmental. However, a common alternative definition does not require that the behavior takes place for environmental reasons. This results in a lack of coherence in marketing literature aiming to profile the green consumer, which is one of its main aims [23].

For the concept of ethical consumption, I1 and I2 on Table 3 provide two different definitions whose foci are rather inconsistent. The former provides a perspective that is concerned with general concepts of ethics and does not specify a single type of behavior, instead it talks about "consumption activities" [37] (p. 507) in general. On the other hand, the latter does specify that the behavior of interest is "the purchase of a product" [38] (p. 512) and further requires that the purchasing decision is made by the consumer "freely" [38] (p. 512). These insights, therefore, exhibit incoherence in their foci which results in confusion and difficulties in integrating conclusions from different studies. The final point that the present comparison aims to illustrate is the overlap between ethical consumption with sustainable and green consumption. In particular, given that sustainability constitutes a major ethical issue [126], sustainable (and green) consumption can be understood as a subset of ethical consumption, since I2 defines ethical consumption as purchasing behavior "that concerns a certain ethical issue" [38] (p. 512), making the overlap apparent.

The fourth and final row of Table 3 shows how the terms "responsible" and "sustainable" consumption are used interchangeably, by the United Nations, to mean the same thing. This is illustrated by the 12th SDG goal having the title of "Responsible consumption and production" [121] (p. 6) as well as the alternative one "Sustainable consumption and production". In conclusion, all the research streams discussed throughout this section are potentially useful for one another. Their constituent pieces of research often study specific consumption behaviors that can be (correctly) considered ethical, responsible, pro-environmental, green or sustainable, and are often just tagged differently from one study to the next.

The CE invites modes of consumption which fit within the research streams discussed throughout this section, as made apparent directly from the goal-oriented part of the concept's definition. However, it is important to recognize that the CE also motivates

consumption and economic practices that are significantly different from those studied in existing research within the aforementioned literature. In particular, an inspection of the material-flow-strategies that characterize the CE through a consumer–cultural lens uncovers initiatives such as Product-as-Service-Systems (PSS), where consumers may be expected to sacrifice ownership of products and purchase the use of the product instead (i.e., use oriented PSS) or where ownership of the product is transferred to the consumer and the provider offers services aiming to extend the useful lifetime of the product over a period of time (i.e., product oriented PSS) (see [39]). A further example that is worth mentioning is collaborative consumption, where consumer-to-consumer business models arise, aiming to maximize the utility of products across consumers, hence reducing the overall resource consumption needed to fill consumer needs in the market (see [40]). Finally, the inter- and trans-disciplinary nature of the CE has implications for consumers even at the product design stage, since it advocates for stakeholder involvement and design processes directed at the better fulfillment of consumer needs (see [41]). While all these implications are very relevant, research exploring them is currently very scarce in comparison to the number of studies that investigate issues like consumption drivers or perceptions, coherent with the extant sustainable consumption (and similar) literature(s) [17]. Therefore, since there is little that a critical appraisal of the limited extant literature can say about research on these alternative practices, the present review focuses on uncovering shortcomings and common misconceptions of existing research that draws from typical practices in sustainable and ethical consumption research, for the context of a CE.

In sum, although the CE has yet to be properly conceptualized in the literature, recent efforts by [5] have yielded a carefully constructed definition from which the present research extracts its core understanding of consumption in the CE. This leads to the conclusion that sustainable consumption is a micro-level basis of the CE concept. Moreover, sustainable consumption has been previously conceptualized under many terms (e.g., Table 3). Therefore, it is important to draw from all the associated literatures in order to successfully identify all previous work relevant to consumption in the CE.

## 4. The Reduce Strategy in Consumption: A Necessary Step towards Sustainability

Consumption patterns implied by the current economic context are unsustainable [1,2]. As demonstrated in Section 1, the current free-market economic systems require that consumption constantly increases in order to sustain their fundamental premises of the economy and full employment. However, what does it mean for this economic system and its encouragement of a consumerist culture to be unsustainable? There are several empirical observations that illustrate this:

- "Earth overshoot day": The day, every year, on which that year's renewable natural resources have been exhausted. After that day, the world population is consuming from the stock of resources which is not replenished naturally. Simply put, this day marks the point at which the planet's yearly natural resource regeneration capacities have been exceeded by human–economic activities. The Global Footprint Network [127] calculates this every year. Since 1980, "Earth overshoot day" arrives earlier each year.
- "Planetary boundaries": These define boundaries on nine separate processes [3,4] which are set based on the planet's regeneration capacities, the conditions necessary for human life and human-originated perturbations on the ES. At least three of these (namely the rate of biodiversity loss, human interference with the nitrogen cycle and climate change) had already been exceeded to a worrying degree in 2008 [3] and, since then, this has only gotten worse [4].
- Soil erosion and degradation: Pimentel [42] shows that degradation and erosion of soil is currently taking place somewhere between 10 and 40 times faster than the rate at which the soil can naturally be replenished. This results in the land becoming unproductive and by 2006, 30% of the worlds workable land had suffered in this way.

- Non-renewable resources: By 2008, of the non-renewable resources currently necessary to support the technologically advanced industrial society, over 70% had already been deemed globally scarce [43].

Although the above list is not exhaustive, it serves the desired purpose of supporting and illustrating the idea that the consumption implied by the current free market economic context is, in fact, unsustainable. This is further argued below.

It is natural to ask how the problem of unsustainability can be tackled and what the impact of tackling it may be for society, the environment and ourselves as individuals. In order to address this, it is necessary to first understand the source(s) of the problem. Three main factors are identified in [2,44] to be significant in determining the impact that human–economic activity has on the environment: global population size, people's income levels (i.e., their capacity to consume products and services in society; affluence) and technology. The interaction between these and their potential effects on sustainability are discussed below.

Global population is constantly increasing due to fertility increases in developing countries and life-expectancy increases. The average age of the population is set to increase as life-expectancy increases to surpass the life-expectancy at birth which remains effectively constant [29]. Furthermore, [45] provides data depicting the possible sustainable population sizes for different per-capita amounts of "biocapacity" use (in global hectares). The biocapacity use per person is directly related to the per capita income and is, therefore, also a measure of how consumption affects the sustainability of the system. Their results show that with the global average per capita income of 2005 (i.e., at 2.7 global hectares of biocapacity/person), the maximum sustainable population size would have been of 5 billion (and not 7 billion as it was). These data support the affluence [45], or income factor, as being even more salient than population size [29]. Finally, the dominant perspective that technology will be able to solve ecological problems through innovation and the right policies [29] runs the risk of becoming counterproductive given the implications of Jevon's paradox [28] and other rebound effects [46]. More precisely, these innovations can result in increased efficiency allowing for enhanced production that can, in turn, drive further over-consumption of resources. Moreover, even products or services exhibiting technological innovations that mitigate environmental/social impacts cannot succeed unless the end consumers accept and adopt them. To summarize, although population and technological advances can prove to be very helpful in achieving sustainable development, they are not sufficient in themselves if the biocapacity consumed per capita (i.e., total consumption) is not addressed.

In spite of that, to date, the most prominent attempts to drive consumption patterns towards more sustainable, focus on the production of less impactful or more efficient product or service alternatives (i.e., green consumerism). Although these strategies can contribute to the solution, they are not enough. In fact, as [27] explains, these attempts may have rather served as a "green card" to allow governments to seemingly address sustainability issues while still protecting a system that incentivizes unsustainable consumerist cultures and consumption patterns. Therefore, attempts for sustainability that do not explore deeper systemic changes fall short of acknowledging the fact that the free market economy, being solely driven by continuous economic growth (as currently understood), unavoidably results in unsustainable production and consumption patterns [47]. The implications of this insight are of great importance. Making consumption sustainable requires making changes to the economic system, the infrastructures and institutions, the power relationships and the dominant lifestyles and consumption culture [30].

From a consumption point of view, this means a cultural transition from the current state of consumerism to one that values more sustainable lifestyles [48]. The idea of sufficiency [49] can play a key role in enabling the attainment of sustainable development. Sufficiency is concerned with the quantity ("how much") as opposed to the quality of what is consumed (e.g., green consumerism). Therefore, the central question to address is that of: How much is enough? Answering this requires finding the balance between

individual/social well-being and ecological or environmental sustainability. Additionally, since these ideas fundamentally challenge the current economic systems, finding this balance is not the only aspect of sufficiency that policymakers and the public perceive as challenging (at the very least).

Among the most prominent is the belief that achieving certain levels of well-being requires material possessions, which is not supported by scientific evidence. Instead, research shows that beyond a certain threshold, material possessions (or energy use) and well-being become decoupled from one another [47,50]. In fact, [51] also determined that the increase in subjective well-being due to income increases becomes non-significant beyond some level of income. For some time already, well-being and economic growth have been decoupled in developed countries [47,52,53]. Moreover, unsustainable consumption patterns have been linked to increases in inequity levels and vice versa [54,55]. In other words, the well-being that can be achieved through material possessions, monetary gains or economic growth (as currently understood), is limited. This supports the existence of some rate or level of resource consumption that is enough, or sufficient.

A culture of sufficiency is also commonly argued against through the idea that the human being is naturally greedy. This would imply that consumerist cultures stem from deeper traits that are inherently human, making them difficult and unnatural to change. However, there is ample research suggesting that this is not the case and that consumerist culture is not a natural occurrence [29]. In the case of the USA, for instance, it was the result of careful design and consequent interventions by think-tanks, government and trade unions, and the benefit of big businesses was its main purpose [48,56,57].

The issues discussed above consider the formal definition of sufficiency (for the purpose of sustainability), its practicality and ethical aspects of the cultural transition. These are conceptual issues attached to a cultural state of sufficiency. Conversely, several practical visions of what a sustainable future that values a culture of sufficiency might look like exist such as, the Venus project [58], which proposes a resource-based economy, and the Sustaining Partnerships to Enhance Rural Enterprise and Agribusiness Development (SPREAD) Sustainable Lifestyles 2050 project [59]. However, the main challenges of the transition are attached to the development of strategies and policies aiming for such a cultural state [29]. Given that the unsustainability of production and consumption is a natural symptom of the underlying system (which is based on a paradigm of endless economic growth), approaches targeting individual and voluntary change will not suffice, despite their utility [30,47]. Given that human behavior is affected directly (internal process that derives in a certain behavior) and indirectly (an external stimulus affecting an internal process in decision making and, consequently, behavior) [60] the previous observations can be summarized as follows: Attempting to shift consumption towards being more sustainable by only affecting consumer behavior directly is not possible. Successfully achieving this also requires the design of an appropriate context (or system) capable of driving the right behavioral change (i.e., indirectly). This insight, given that consumers are embedded in society, through norms and regulation, illustrates the need to consider institutions.

Institutional research has long emphasized the understanding of institutions as structural forces that induce stability [61–63]. By distinguishing what is legitimate from what is not (i.e., by providing meaning), they facilitate the prediction of other agents' behavior, making social interactions more stable and meaningful [64]. In other words, institutions influence agent-level behavior eventually inducing homogeneity (stability) in the system. However, this exercise of legitimization also occurs from the bottom-up, such that institutional agents (organizations, individuals etc.) can also shape the institutions. These two ways of legitimization depict what neo-institutional theory terms "deterministic" (static; top-down) and "strategic" (dynamic; bottom-up) stages of the organizational field [65]. Moreover, institutional order lies on three fundamental pillars: regulative, normative and cultural–cognitive [66]. The regulative pillar gathers institutional forces emergent from rules that are set, monitored for and sanctioned against at the macro level of societal organization. The normative pillar includes prescriptive elements of norms, standards and values

that drive institutional stability [65]. Finally, the cultural–cognitive pillar is related to the cognitive structures through which meaning is created and the social knowledge shared across agents in a given institutional context [66]. Therefore, these three pillars support every consumption behavior, and so, changes in any of the three can influence behavior [65]. This illustrates two fundamental building blocks of neo-institutional theory [66], namely, the pillars of institutional organization and legitimacy as a necessary condition for the stability of the institution.

The implications of these tenets are extremely relevant to sustainable consumption. First, they require that institutions are legitimized at all three levels (pillars) in order to stabilize. Second, legitimacy both determines the rules under which agents operate and is also a resource through which agents can drive institutional change [65]. In [67], the author explains that institutions shape agents' cognition, therefore guiding their social behavior, one of which is consumption (this describes an indirect effect on behavior). As a result, taking a neo-institutional view on consumer behavior can provide insights on the external and internal factors guiding or hindering SCB.

Existing groups and movements, such as the voluntary simplicity movement and others [68–70], that are attempting to support a shift in consumer culture through initiatives that target individual voluntary change (i.e., internal shocks to the system), are not large enough in scope and reach to drive systemic change alone [29]. Therefore, the right legislation (effectively an external shock) must be put in place to support and guide this change at a large enough scale. That being said, initiatives such as the Transition Network [71] aiming to facilitate the organized action of communities locally, sparking pro-environmental entrepreneurship, could potentially have a significant system-wide effect through practically exemplifying the future in a tangible manner for policymakers and society.

The need for evidence-based policy capable of driving behavioral change is recognized among policymakers [60,72,73]. This implies the need to elucidate how macro-level strategies can drive the consumer–cultural change required to achieve sustainable development. Some research streams work to elucidate governance opportunities that can drive change, most notably, system innovation research [74] and evolutionary economics [75]. However, since individual agents' social behavior arises partly due to the institutional context shaping and conditioning their cognition [67], cognitive–behavioral effects must also be understood in order to complete the picture. That is, understanding the individual agent (consumer, household, organization, etc.) level from a cognitive–social point of view, like in research streams such as psychology [76,77], microeconomics [78–80] and marketing [81,82] is vital in ensuring the success of policies aiming to drive the right consumer–cultural change [16].

In sum, resource consumption and resulting shocks on the ES's boundaries are taking place at unsustainable rates. Additionally, the consumer–cultural changes required for attaining sustainable consumption levels call for a neo-institutional perspective that recognizes the need for sustainability to be legitimized at the regulative, normative and cultural–cognitive pillars. Through this lens, it is recognized that institutional agents can also affect institutional structures. However, in the context of sustainability, bottom-up initiatives have proven to be too weak to drive system-wide change. This calls for the recognition of governance's responsibility in laying the right environment and regulatory framework to shift consumer behavior towards more sustainable. Finally, developing the right policies and context also requires an understanding of the cognitive–behavioral side of consumption, where the streams of research reviewed in more detail in the following section usually operate.

## 5. The Intention–Behavior Gap in Sustainable Consumption: Establishing the State-of-the-Art

Sustainable consumer behavior is defined as consumption behavior that is "oriented toward sustainable development" [32] (p. 197). In their work, [83] review existing methods for the measurement of pro-environmental, or sustainable, consumer behavior. Within the self-reported measures of SCB that the authors identify, there are several approaches that differ in: (a) their focus on varying behavioral properties, (b) time frame (i.e., present,

specified past or unspecified past) and (c) the generality of the target behaviors. Notably, the review identifies three main types of scales that have been commonly used. First, several measures of SCB have been designed to capture the participant's tendency to engage, or not, in PECB across varying domains. These are essentially behavior-specific pre-dispositions to behave pro-environmentally in some domain pertaining to SCB, or many domains simultaneously. Second, there are "diary procedures" where the participants are required to report on their SCB more than once, hence keeping a record of their target SCB over some established time (e.g., [84]). Finally, there are measures of individual environmental footprint (EF) that rely on the participants' self-reports of their habitual behaviors, focusing on key behaviors responsible for the largest portion of an average individuals' EF [85]. These measures of EF are particularly relevant given the discussion throughout this review about the importance of addressing the goal-side of the CE's definition. Additionally, a combination of the diary procedures method with the impact-based EF measures of SCB may offer interesting avenues for increased precision in the measurement.

Measures that quantify PECB through the calculation of individuals' EF may first appear to measure something different from behavior. However, they are the only instrument that explicitly considers the impacts associated with its target behaviors. This is essential since, by definition, it is the impacts associated with the different behaviors that determine their sustainability, or lack thereof. Consequently, issues of sustainability require an analysis from a consequentialist perspective. Meaning that it does not suffice to (deontologically) set a number of rules about the quality (type) of consumption if said rules allow for aggregate levels of consumption that are not sustainable (i.e., whose impacts hinder their sustainability). In other words, SCB can only be properly defined and measured if the consequences (or impacts) of said behavior are taken into account. Moreover, impact-based measures of sustainable consumer behavior offer a particularly convenient framework for assessing consumer involvement in the most highly preferred of the CE-initiatives, namely the "reduce" pseudo-loop.

As discussed in Section 4, SCB is not only concerned with the quality, but also the quantity, of consumption. Furthermore, ethical consumption addresses consumption practices that entail ethical considerations. The consumption patterns that the current free-market capitalist systems require carry serious ethical weight given the associated social, economic and environmental impacts and risks. Therefore, ethical consumption is necessarily concerned with the issue of sustainable consumption. As a result, ethical consumption, like SCB, must also consider the quantity of consumption [128] on top of its qualitative features (e.g., green consumerism) [129]. This is not surprising given the overlap between ethical consumption and SCB illustrated in Section 3.

Abundant empirical research reports positive attitudes, preferences and BI towards more sustainable [130–134] and ethical [100,135–137] products, suggesting that, on top of associated environmental and social benefits, the sustainable/ethical aspects of products can be a competitive differentiating factor. As a result, the consumer behavior literature in marketing, and the consumer psychology literature exhibit a growing interest in SCB, ethical consumption and similar concepts [138–141]. However, most extant research on SCB and consumption in the CE currently relies on self-reported data, often in hypothetical settings [17,86], posing significant limitations to the reliability of the results and the conclusions that can be drawn about actual behavior [86,87]. Moreover, the common reliance on models built around the idea that BI is a strong predictor of behavior [101], characteristic of the widely favored Theory of Planned Behavior (TPB) [17], means that the research has focused mostly on understanding the formation of BI [24]. Therefore, these ignore empirical claims from the wider social psychology and consumer behavior literature that BI may not generally be assumed to translate into behavior [88,102]. This has led to a poor understanding of the translation of BI into behavior in ethical contexts [100].

Not surprisingly, abundant research in the SCB literature (and other relevant literature, identified in Section 3) agrees on the existence of an IBG. These terms refer to the following widely reported phenomenon: *There exists a strongly significant mismatch between consumers'*

*stated attitudes and/or BI, and their actual behavior* [25]. For example, most Europeans report that they are aware of the unsustainability of current consumption patterns and the importance of resource effectiveness in overcoming it [142]. They also claim to engage in waste management practices and to be willing to engage with new business models [26]. However, these claims do not match observations of the real world [16]. In [143] (p. 1), the authors label the "inconsistency between what people say and what they actually do" (i.e., the IBG) as "the most consistent finding" within their literature review. Although not a new problem and having been reported in numerous contexts of individual ethical behavior, the IBG phenomenon remains poorly understood [100,144]. The lack of studies capturing actual behavior, rather than self-reported data, and the lack of heterogeneity in the models employed to study SCB are potential sources of the gap [89]. In sum, given the limitations that the IBG poses on the reliability of extant research, understanding the IBG constitutes one of the main challenges to be addressed by the SCB literature, as reported by [90]. A more in-depth account of existing perspectives on the IBG follows.

*Perspectives on the Intention–Behavior Gap in Sustainable Consumption*

Within the SCB literature concerned with the IBG, two overall perspectives can be identified [91–93]. The first, proponents of which the authors [91] name modelers, attempts to find non-methodological explanations to the IBG. In other words, although typical survey methodologies are recognized to be partially responsible for the gap, other factors are considered to be more significant. Consequently, modelers attempt to identify factors and processes that may hinder SCB. Among the most common, are explanations of the IBG such as barriers to SCB such as price premiums overtaking consumers' willingness to pay extra [103,104], the unavailability of sustainable/green alternatives [104,105] and lower perceived quality [106]. Further common explanations of the gap include situational factors, such as citizen–consumer role conflicts [107], and information-based reasons like lack of knowledge or trust regarding some sustainable product/service [108] and information overload, typically present in a market setting [109].

Although the identification of barriers uncovers part of the puzzle, it remains to explore the processes through which these barriers influence the BI–behavior relationship. Consequently, conceptual propositions have emerged that integrate several factors to model mechanisms of influence on the BI–behavior relationship. Most notably, [25] model implementation intentions (see [110]) to positively mediate the BI–behavior relationship and introduce actual behavioral control (see [111]) and situational context (see [112]) as two moderators of the relationship. Moreover, they model the strength of the BI–behavior relationship as being positively affected by consumers' control over the behavioral experience (actual behavioral control) and potentially influenced by the context in which the behavior takes place (situational context). In [113], the authors conduct an empirical test of the model in [25] and, while they find implementation intentions to positively mediate the BI–behavior relationship, as expected, they find behavioral control to just mildly moderate the relationship. Furthermore, situational context turned out to not have a significant effect, leaving the model only partially supported. The authors also incorporate a further construct to the model developed in [25], in order to capture consumer involvement in environmental causes. However, its hypothesized positive moderation effect on the BI–behavior relationship was only mildly supported by their results.

Another important example lies on the analytical framework of cognitive dissonance. Qualitative exploratory research in this line has found the lack of cognitive dissonance to be a significant perpetuator of the IBG [114–117]. While different in goals and foci, these studies agree that consumers who have sustainable BI tend to utilize strategies to neutralize psychological discomfort that results from the incoherence between their behavior and their goals. In [114], the authors focus solely on identifying normalization strategies that green consumers use to lower the dissonant cognition emergent from travelling by plane, since it is highly impactful, while in [116] they focus on conceptualizing the "conscious consumer". The authors in [115] investigate and conceptualize the role of emotions and

the prevalence of incongruent behavior in the context of ethical consumption and in [117] the focus is on incorporating a "neutralization" construct into the TPB framework, in the context of fair-trade product purchase. More recently [118], a model was developed and tested that focuses on understanding the role of psychological discomfort (originating from incongruent behaviors) in behavioral change in the context of meat consumption. The authors, like the aforementioned exploratory studies, include neutralization-like constructs in the form of "trivialization" and "detribalization" of dissonant information. Their findings suggest that, while psychological discomfort does positively affect motivation to change the dissonant past behavior, this tends to fade away with time. This suggests that the IBG may be enlarged over time through neutralization-like strategies.

The second point of view, held by the methodologists [91], views the gap as mainly a consequence of methodological biases. As discussed earlier, the preferred self-reported survey methodologies are considered to limit the reliability of results [86,87]. Under this category are understandings of the IBG that consider factors such as exaggeration and social desirability bias [87], whereby respondents are compelled to respond according to what they believe to be socially desirable, and not their individual thoughts and beliefs. Moreover, the effect of social desirability has been found to be particularly salient in an ethical research context [144]. Another significant IBG factor identified by the methodologist perspective on the IBG is the hypothetical bias, whereby responses to hypothetical questions may be biased by the respondents' imprecision in predicting their behavior by just imagining a given scenario [88,94,95]. On the other hand, the authors of [96] conduct a study to identify the significance of the effect that such methodological biases have on the reliability of associated results and find that it is not typically significant. Similarly, [94] find the differences in adaptation levels of the consumer between the hypothetical and real market settings to be more significant than the hypothetical bias. However, such accounts are rare in comparison to the converse idea that methodological biases are in fact significantly limiting [86,95,96].

While methodological constraints are potentially significant [86], most research still uses self-reported data [24] without addressing the associated limitations (e.g., social desirability, self-presentational biases and self-reporting benefits exaggeration [87]). Attempts to overcome these methodological limitations and explore the IBG have followed experimental routes. On one hand, there are field experiments with point-of-sale observations, particularly in contexts of organic or fair-trade grocery purchasing [91,94,107,145]. Although field experiments offer significant qualitative insight into the moment of purchase, quantitative observations are limited by noise that cannot be controlled for. On the other hand, while experiments in a laboratory setting have also been conducted for similar markets [91,97,98], these do not in general meet incentive compatibility requirements and typically involve hypothetical settings with intangible consequences to participants' choices in the experiments. Namely, incentive compatibility requires (1) that choices are made between strictly less than three options and (2) that the respondents must care about the problem raised by the survey and that they believe that their choices will have some real impact for them [99,146]. These conditions are necessary in order to ensure that behavioral data collected are in fact representative of actual behavior [99,146]. Therefore, the observation/measurement of real behavior in a controlled experimental setting for the exploration of the IBG in sustainable consumption is currently missing from the literature [147]. For this reason, for example the authors in [97], call for replication of their results using real monetary incentives, allowing the otherwise persisting methodological limitations of using hypothetical settings and non-compatible incentives to be addressed and overcome.

There are other, more marginal exploratory attempts to conceptually explain the IBG that cannot be classified as taking a methodologist or modeler perspective. For example, the authors of [148] explore the idea that the IBG may be the result of the misalignment between consumers' perception of the impacts of a sustainable product alternative and the actual sustainability of the product as determined through life-cycle assessment. However,

the IBG is not the focus of their study and therefore remains an unexplored idea. Moreover, as both methodologist and modeler perspectives suggest, this is unlikely to be the only, or most, significant source of the IBG.

In sum, there are currently two salient perspectives in understanding IBG in SCB, the modeler and the methodologist. In the former strand, that understands the IBG as originating from the models used, there are two conceptual propositions that are gaining traction that relate to the concepts of implementation intentions (i.e., specific plans to act upon one's intentions) and cognitive dissonance (i.e., the dissonance that emerges from not acting as intended), respectively. However, these have mostly sprouted as exploratory studies and are far from widely validated. Therefore, the modeler perspective still requires inputs in the form of both, conceptual models and empirical tests. While a large majority of research within the methodologist perspective advocates for the significance of methodological biases, there have been few attempts to overcome and explore the effects of methodological limitations on the IBG by gathering data representative of actual behavior. The field and laboratory experiments that have been conducted focus on purchasing behavior in the context of organic or fair-trade groceries. Furthermore, extant experimental attempts in the field do not allow for the proper control of the experimental setting and extant experimental attempts in the laboratory fail to meet incentive compatibility requirements. Therefore, the present review calls for innovation towards abstract experimental approaches more common to the domains of experimental economics and psychology. Both methodological care and conceptual innovation are necessary in order to contribute to understanding the mechanisms behind SCB.

## 6. Discussion

The present critical review identified several key points that can be used as the ground on which to build a much-needed bridge between existing research exploring SCB and related concepts (see Section 3), and research on consumption in the CE. In doing so, the review contributes to the sustainability and CE literature by addressing the issue of the critical characterization of consumption in the CE and its relation to existing conceptions of and research on SCB. The review also contributes to the aforementioned literatures by establishing a connection between the growing discourse in the sustainable consumption literature, about the need for a shift away from consumerist cultures, and the incorporation of this idea as part of the reduce strategy within the CE framework. Finally, our review further contributes to the aforementioned bodies of literature, as well as the economics, psychology and marketing literature, and their boundaries, by gathering and organizing research on the IBG in contexts of sustainability-relevant consumption and decision making. In this way, we open the floor for all these disciplines to employ their methodological and theoretical strengths to further understand the IBG, and consumption in contexts relevant to sustainability considerations. This is a particularly relevant contribution in the context of CE research since the concept itself advocates for interdisciplinarity.

First, two core defining elements or pillars that are necessary in order to build the CE concept were extracted, among others, directly from [5]'s work and definition. These are: (1) There exists a hierarchy of CE-strategies stemming from the nR-framework, meaning that some strategies are preferred to others. In particular, the "reduce" strategy (or loop) is argued to be of paramount importance in enabling the success of the transition to a CE from the consumption side (see Section 4). (2) The definition of the CE consists of two separable parts: its goal, which can be summarized as the attainment of sustainable development; and its strategies for achieving the goal. Therefore, only the combination of the CE-strategies together with their contribution towards sustainable development can be considered part of the CE.

While the CE can be considered a fairly new concept, consumer behavior that is relevant to the attainment of sustainable development through often similar strategies has been studied to a reasonable extent under a number of names. However, these streams of research (ethical, responsible, pro-environmental, green and sustainable consumption

or consumer behavior) are characterized by a lack of heterogeneity in the theories used, focusing primarily on the formation of intentions, often falling short of really studying behavior. Moreover, the typical use of survey methods and self-reported data poses a further problem for these research streams. In particular, these issues give rise to a phenomenon known as the intentions–behavior gap, whereby consumers report being environmentally conscious and intending to behave sustainably but fail to act accordingly. Putting the IBG at the center of the discussion opens up interesting avenues for extending this type of research to the CE in an adaptive manner. Namely, it calls for innovation in the use of testable theoretical frameworks to understand consumption behavior and lifestyles or cultural elements of their behavior, relevant to the transition towards the CE. Moreover, the discussion calls for particular methodological care when working with self-reported measures of behavior, through using impact-based instruments for instance. Finally, measuring behavior in laboratory–experiment settings where particular care is given to the design, in order to ensure incentive compatibility, may offer a particularly interesting solution to the methodological shortcomings of these streams of research exploring SCB.

Key managerial implications of the present review include the recognition that given the unprecedented diffusion of the CE concept across societal stakeholders (industry, academia and government), there is a growing need for firms to adapt to its tenets. What is more, in the current context where policy and academic research on the CE paradigm are still young and the transition towards more sustainable modes of consumption and production, while inevitable and fast-moving, is still at an early stage; managers can benefit greatly from the early adoption of CE-strategies by creating precedents and knowledge whose value will eventually unavoidably increase. Furthermore, consumers' growing care and knowledge about environmental and sustainability-related issues that the world collectively faces, calls for transparency in marketing of green attributes of products. In addition, it calls for the divergence from product or service design and business strategy development that is detached from consumers and aimed at the creation of value through marketing, and the convergence towards design processes that develop a real understanding of consumer needs through their direct involvement.

Reflecting upon our discussion of the important role held by a culture of sufficiency in a shift towards sustainable development (see Section 4), managers should seek avenues through which to contribute to the creation of such a culture that can be financially sustainable. For example, through marketing of the benefits of durability, repairability and recoverability and even the incorporation of repair, remanufacture and recovery services into their business models. Finally, when conducting market research on "green" consumer segments, firms should be weary of results obtained through self-reported data and typical methods employed currently, since intentions and self-reports may significantly misrepresent subsequent consumer choices and behavior (see Section 5), and favor the use of field experiments and case study methods.

There are limitations to this review which need acknowledging. A first arises from our chosen review approach, since critical reviews are known to not meet the levels of systematicity present in other, more structured approaches [120]. Recognizing this, our review employed literature search methods drawn from typical practices in systematic literature reviews in order to minimize the issue of subjectivity by reporting on the snowball mapping and key-word strategies employed for the identification of sources and the subsequent inclusion criteria employed. A further limitation worth discussing may also be understood as originating in the choice of the critical analysis approach. Due to the scope and reach of critical reviews, our work does not explore new modes of consumption and other implications of the CE for consumers (i.e., PSS, collaborative consumption, stakeholder involvement in product design etc.) in detail. While we do identify them as relevant to the characterization of consumption in the CE, the review's goal of bridging extant research streams with new considerations of consumption in the CE, combined with the small number of studies that currently explore these issues, makes these unsuitable for a critical appraisal of the limited research currently available. Therefore, there is

a need for future research that explores these new implications of the CE and modes of consumption, which could still benefit from our review's findings and discussion. Additionally, while the focus of our review lies on the consumption-related aspects of the citizen–society relationship, and its connection to the environment, other interesting implications may arise for other literatures. To name one, the cognitive–cultural changes necessary for the success of the CE, as well as the innovation in business models (and arguably societies as a whole) implied, could have effects on the labor market. For instance, by making citizens more willing to work for sustainability-principle abiding, circular firms or organizations.

Further research on consumption in the CE should attempt to recognize and address the two defining pillars identified herein (see Section 3 and Table 1) offering justification for the choice of study based on the hierarchy of CE-strategies (or material-flows) and the impacts associated with the studies' subject. Moreover, studies of the cultural–cognitive elements associated with attitudes towards and adoption of sufficiency-based lifestyles are particularly relevant. In this respect, considerations that employ an institutional lens to explore how the micro, meso and macro institutional levels may give rise to such a state of equilibrium, where sufficiency is valued, are necessary. Additionally, how the aforementioned topics relate to consumers' engagement in alternative modes of consumption, involvement at the product-design stage or adoption of other circular strategies would be of particular interest to management research but also for research in economics or psychology. Finally, there is a need to explore the impact mitigation potential, at the appropriate scales, of the cultural shift toward sufficiency, of the alternative modes of consumption and of other strategies for sustainability implied by the CE.

The IBG (see Section 5 and Table 1) poses a significant limitation to research following the lead of the extant SCB, and related, literatures. More importantly, however, it uncovers both methodological and conceptual issues that need to be addressed by future research. Namely, research employing self-reported data collection methods should attempt to use instruments that are less susceptible to social desirability bias and similar problems, such as impact-based measures or diary procedures. Particularly, innovation in methodologies, towards experiment procedures that meet incentive compatibility criteria when considering consumption in the CE or SCB as a whole and more abstract experiments, may offer the best avenue with regard to addressing this issue. In addition, attempts to measure and record the effect of such biases are necessary, in order to better understand the significance of the methodological part of the problem. Further, new conceptual models and empirical testing of existing understandings of the IBG, and attempts to overcome it, are necessary in order to better understand the IBG, but also SCB as a whole.

**Author Contributions:** Conceptualization, D.G.G., E.K., E.V. and A.S.; methodology, D.G.G.; formal analysis, D.G.G., E.K., E.V. and A.S.; writing—original draft preparation, D.G.G.; writing—review and editing, D.G.G., E.K., E.V. and A.S.; visualization, D.G.G.; supervision, E.K., E.V. and A.S. All authors have read and agreed to the published version of the manuscript.

**Funding:** This research has received funding from the European Union's Horizon 2020 research and innovation programme under the Marie Skłodowska-Curie Innovative Training Networks (H2020-MSCA-ITN-2018) scheme, grant agreement number 814247 (ReTraCE project).

**Institutional Review Board Statement:** Not applicable.

**Informed Consent Statement:** Not applicable.

**Data Availability Statement:** No databases were employed in the present review. All results can be readily contrasted against the literature sources in the reference list.

**Acknowledgments:** We thank seminar participants at The University of Sheffield (UK), South-East European Research Centre (Greece) and ReTraCE (H2020-MSCA-ITN-2018; grant agreement number 814247) events participants for offering their valuable attention and feedback. We also thank Fraser Mcleay (Specific examiner; University of Sheffield) and George Eleftherakis (Exam chair, SEERC) for providing Mr Georgantzis Garcia and supervisory team with a very stimulating confirmation review

process and discussion which gave impetus to further considerations and subsequent improvements in this work.

**Conflicts of Interest:** The authors declare no conflict of interest. The funders had no role in the design of the study; in the collection, analyses, or interpretation of data; in the writing of the manuscript, or in the decision to publish the results.

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
