# Peer review of "Consumption in the Circular Economy: Learning from Our Mistakes"

_sustainability, doi:10.3390/su13020601_

Round 1

Reviewer 1 Report

Summary: The current manuscript reviews the literature, by means of snowball mapping and a systematic key-word searches, aiming to elucidate the fundamental elements that should characterize consumption in a Circular Economy (CE). The results extract two strategies to be held: The hierarchical nature of circular strategies; and (2) The inadequacy of defining the CE only through its loops or strategies 19 without considering its goal of attaining sustainable development. Overall, this job has important theoretical contribution for the development of working strategies aiming to increase sustainability among consumer behavior. More broadly, this work fits nicely with emerging work in sustainability testing the most efficient methods for bolstering environmentally firms behaviors. Therefore, it adheres to the journal’s standards. While interesting, there are remarkable issues that limit its contribution to the literature in sustainability. Broadly speaking, remarkable improvements in methodological procedures, bibliography indications and discussions are needed to fully evaluate the validity of the results.

Below, my questions are arranged by the order in which they were motivated

while reading the manuscript:  1. Abstract: The authors are good storytellers.

However, a brief specification of the method used for the systematic review is

needed.

2. Introduction:  While I do agree with the research gap the authors outline,

the authors do need to update their references and include more works talking

about the importance of considering the key role of consumers in the CE.

For example: what is the difference between your work and that by Sing and

Ordóñez (2016), Testa et al. (2020) or Mugge (2018). In other words, you

should develop a clearer justification of the research gap you intend to fulfill. 

3. There is not a clear Research Questions development neither statement. 

4. Method: If this study attempts to be published, the authors must better

specify all the methodology behind a review or bibliometric analysis.

This is my main concern. Specifically:-        Which procedures have the authors

followed to include such studies?-        Which were the database that the

authors used? Range of years?-        Why have you selected such topics? -  

      Which tool have the authors used to make such a combination of articles.

If readers do not have answers to these questions, the results can be highly

subjective and out of the rigour of this journal. Please, see some papers such

as Muñoz-Leiva et al. (2012), Rodríguez-Lopez et al. (2019), Casado-Aranda

et al., (2020), or Bastdias-Manzano et al. (2020) for some insight. 

5. The authors would need to more deeply explain your contribution to the

literature in sustainability research, but also to those studies implemented in

the field of working/labor research.  Of course, at this regard, it would be of

high interest including research in the field of environmental retailing and

practices. 6. Highly simplistic and obvious your limitations, managerial

implications and further research.  Best of luck as you continue with this

research.

  1.  

Singh, J.; Ordoñez, I. Resource recovery from post-consumer waste: important lessons for the upcoming circular economy. Journal of Cleaner Production 2016, 134, 342–353, doi:10.1016/j.jclepro.2015.12.020.

Testa, F., Iovino, R., & Iraldo, F. (2020). The circular economy and consumer behaviour: The mediating role of information seeking in buying circular packaging. Business Strategy and the Environment.

Muñoz-Leiva, F., Viedma-del-Jesús, M. I., Sánchez-Fernández, J., & López-Herrera, A. G. (2012). An application of co-word analysis and bibliometric maps for detecting the most highlighting themes in the consumer behaviour research from a longitudinal perspective. Quality & Quantity, 46(4), 1077–1095. https://doi.org/10.1007/s11135-011-9565-3

Rodríguez-López, M. E., Alcántara-Pilar, J. M., Del Barrio-García, S., & Muñoz-Leiva, F. (2019). A review of restaurant research in the last two decades: A bibliometric analysis. International Journal of Hospitality Management, 102387. https://doi.org/10.1016/j.ijhm.2019.102387

Bastidas-Manzano, A.-B., Sánchez-Fernández, J., & Casado-Aranda, L.-A. (2020). The Past, Present, and Future of Smart Tourism Destinations: A Bibliometric Analysis. Journal of Hospitality & Tourism Research, 1096348020967062. https://doi.org/10.1177/1096348020967062

Casado-Aranda, L.-A., Sánchez-Fernández, J., & Viedma-del-Jesús, M. I. (2020). Analysis of the scientific production of the effect of COVID-19 on the environment: A bibliometric study. Environmental Research, 110416. https://doi.org/10.1016/j.envres.2020.110416

Reviewer 2 Report

The paper is well written. It contains a broad review of the literature and a thorough review and classification work is observed. However, it would be interesting to include a table of groupings of the different aspects of the review carried out, so that it is easier to understand it and detect the gaps in the research detected.

Reviewer 3 Report

Thanks for the manuscript, the topic you decided to address is very relevant the importance of consumption in enabling  CE’s success in terms of consumers’ acceptance and adoption and in attaining sustainable development.   The authors correctly recognise that the consumption side has not been given nearly as much attention as the production side in the CE literature. The goal is to elucidate the fundamental elements that should characterise consumption in a CE through a critical review of existing studies.    The objective is clear, literature review is complete and discussion highlights important points.    My major comment is about Section3. I was expecting also some comments about how the CE will differ from other forms of sustainable consumption, but I noticed that only similarities and overlaps were highlighted. For example, nowhere in this section the authors refers to the fact that the CE will also change modes of consumption which are not expected in other definitions of sustainable/ethical consumption mentioned in the paper (e.g. shift form ownership to access: product as a service). A real CE has the potential to change forms of consumption in a way that can be considerably different from other forms of sustainable or ethical consumption. I think this is important to address. The rest is fine.   Other minor comments:   Section 7 (line 140) is actually Section 6.   The acronym SCB , firstly mentioned in line 137, is never defined. Since it might be of interest for researchers from different fields, I would define it clearly (sustainable consumption behaviour, right?) .   Line 194 - OPTIONAL: I don’t know if it can be useful for the authors or not, but the hierarchy of the CE is very well integrated in the Zero Waste movement (see Zero Waste Alliance), which makes this hierarchy its core pillar.  See if mentioning this can add value to the manuscript.   Line 396-97: I would add the Transition Town movement, very important in terms of behavioural consumption patterns.    Finally, I suggest a thorough review of the manuscript because there are several typos and missing words.

Round 2

Reviewer 3 Report

The authors have properly addressed all the comments. The manuscript has been significantly improved and can be published in Sustainability.